# `baller2vec++`: A Look-Ahead Multi-Entity Transformer For Modeling Coordinated Agents

## Abstract

In many multi-agent spatiotemporal systems, agents operate under the influence of shared, *unobserved* variables (e.g., the play a team is executing in a game of basketball). As a result, the trajectories of the agents are often statistically *dependent* at any given time step; however, almost universally, multi-agent models implicitly assume the agents' trajectories are statistically *independent* at each time step. In this paper, we introduce `baller2vec++`[1], a multi-entity Transformer that can effectively model coordinated agents. Specifically, `baller2vec++` applies a specially designed self-attention mask to a *mixture* of location *and* "look-ahead" trajectory sequences to learn the distributions of statistically dependent agent trajectories. We show that, unlike `baller2vec` (`baller2vec++`'s predecessor), `baller2vec++` can learn to emulate the behavior of perfectly coordinated agents in a simulated toy dataset. Additionally, when modeling the trajectories of professional basketball players, `baller2vec++` outperforms `baller2vec` by a wide margin.

## 1 Introduction and Related Work

Whether it is a team executing a play in a game of basketball, a family navigating to an attraction in a theme park, or friends posting about a birthday party on a social media platform, humans frequently coordinate their behavior in response to shared information. When this coordinating information is unobserved (which is often the case in many machine learning datasets), the individuals' observed behaviors become correlated, i.e., the behavior of one individual at a specific moment contains information about the behavior of another individual *at the same time*. In the context of modeling agent trajectories in multi-agent spatiotemporal systems, this property translates to the trajectories being statistically dependent at each time step. However, nearly all multi-agent spatiotemporal models (e.g., [1–6]) implicitly (through their loss functions) assume the trajectories of the agents at each time step are statistically *independent* given the agents' previous locations (Figure 1).

Zhan et al. [7] explicitly focused on modeling coordinated multi-agent trajectories, using "macro-intents" [8] that are shared across agents to do so. The macro-intents are generated from a separately trained recurrent neural network (RNN) that learns to predict a future, coarse, "stationary" location for each agent at each time step. The macro-intents for all of the agents at a specific time step are concatenated together to form a single, shared, macro-intent variable, which is then provided as input to the trajectories-generating model at that time step. However, similar to the previously mentioned multi-agent trajectory models, the macro-intents model im-

---

[1] All data and code for the paper are available at: <anonymized>.

plicitly assumes the macro-intents for the agents at each time step are statistically independent, i.e., the macro-intent for one agent does not depend on the macro-intents of the other agents.[2]

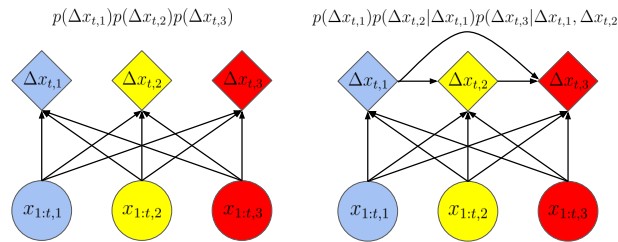

Figure 1: **Left**: most multi-agent systems implicitly assume the trajectories of the agents at each time step ($\Delta x_{t,k}$) are conditionally independent given the agents' previous locations ($x_{1:t,k}$). **Right**: however, the various decompositions of the joint probability of the trajectories, e.g., $p(\Delta x_{t,1})p(\Delta x_{t,2}|\Delta x_{t,1})p(\Delta x_{t,3}|\Delta x_{t,1}\Delta x_{t,2})$ (note, we omit the conditional $x_{1:t,k}$ terms for brevity), suggest more complex statistical dependencies between the agents' trajectories can exist (i.e., the independence assumption is an extremely strong one). Indeed, there are often shared unobserved variables influencing the spatiotemporal behaviors of agents—such as the play that the players on a basketball team are executing, or events occurring in a pedestrian environment—which suggests statistical dependencies between the agents' trajectories are likely.

Further, the trajectories-generating model still implicitly assumes the trajectories of the agents at each time step are independent, which is only true if the shared macro-intent variable perfectly captures all of the unobserved information that could cause the agents' trajectories to be correlated.

Notably, Social-BiGAT [9] *does* partially account for trajectory correlations through a global adversarial loss. Specifically, the trajectories for each agent are separately passed through an encoder RNN, and the final hidden states for each agent are then processed with a graph attention network (GAT) [10]. The output of the GAT is then used as an input to a discriminator that classifies whether or not the input trajectories are real or generated. This global adversarial loss, however, is only a single component of the full Social-BiGAT loss function, and other components of the loss function *do* implicitly make the independence assumption. Further, interestingly, adding the global discriminator to a baseline model only improved the model's performance for one out of six pedestrian datasets.

In this paper, we describe a novel multi-agent spatiotemporal model that integrates information about *concurrent* actions of agents to predict statistically dependent distributions of trajectories. Specifically, we extend the recently introduced multi-entity Transformer `baller2vec` [6] by: (1) augmenting its input with a parallel sequence of "look-ahead" agent trajectories and (2) using a specially designed self-attention mask, which allows our model to exploit the chain rule of probability (Section 3). We find that:

1. `baller2vec++` is an effective learning algorithm for modeling coordinated agents. Unlike `baller2vec`, `baller2vec++` can learn to emulate perfectly coordinated agents from a simulated toy dataset (Section 5.1). Further, `baller2vec++` outperforms `baller2vec` by a wide margin (8.9%) when modeling the trajectories of professional basketball players (Section 5.1).

2. `baller2vec++` makes better predictions when conditioned on concurrent trajectory information from other agents, supporting our proposition that the commonly used independence assumption for agent trajectories is overly strong (Section 5.2).

3. Lastly, the joint probability assigned to a sequence by `baller2vec++` is approximately permutation invariant with respect to the order of the agents, i.e., `baller2vec++` respects the properties of the chain rule (Section 5.3).

---

[2]See the authors' implementation here: `https://github.com/ezhan94/multiagent-programmatic-supervision/blob/a1d9152d4c8a287474953cba093c28fef2a05979/models/macro_vrnn.py#L101`.

## 2 Background

### 2.1 Multi-agent trajectory modelling

Our problem description closely follows Alcorn and Nguyen [6], whom we quote here:

> Let $A = \{1, 2, \ldots, B\}$ be a set indexing $B$ agents and $P = \{p_1, p_2, \ldots, p_K\} \subset A$ be the $K$ agents involved in a particular sequence. [Let] $C_t = \{(x_{t,1}, y_{t,1}), (x_{t,2}, y_{t,2}), \ldots, (x_{t,K}, y_{t,K})\}$ [be] an unordered set of $K$ coordinate pairs such that $(x_{t,k}, y_{t,k})$ are the coordinates for agent $p_k$ at time step $t$. The ordered sequence of sets of coordinates $\mathcal{C} = (C_1, C_2, \ldots, C_T)$, together with $P$, thus defines the trajectories for the $K$ agents over $T$ time steps.

In multi-agent trajectory modeling, the goal is to model a joint probability of the form:

$$p(\Delta x_{t,1}, \Delta x_{t,2}, \ldots, \Delta x_{t,K} | x_{1:t,1}, x_{1:t,2}, \ldots, x_{1:t,K})$$

i.e., the joint probability of the $K$ agents' trajectories $\Delta x_{t,k}$ at time step $t$ given the agents' location histories $x_{1:t,k}$. We note here that the common practice of *simultaneously* predicting the trajectories for all of the agents at a specific time step is not required by theory. Using the chain rule of probability, the joint probability of the agents' trajectories can be factorized as, e.g.:

$$p(\Delta x_{t,1}, \Delta x_{t,2}, \ldots, \Delta x_{t,K}) = p(\Delta x_{t,1})p(\Delta x_{t,2}|\Delta x_{t,1}) \ldots p(\Delta x_{t,K}|\Delta x_{t,1}, \Delta x_{t,2}, \ldots, \Delta x_{t,K-1})$$

where we omit the conditional historical trajectories for brevity. As a result, it is perfectly acceptable to generate trajectories agent-wise, using the previously generated trajectories as additional conditioning information when generating the trajectories for later agents (see Figure 2).

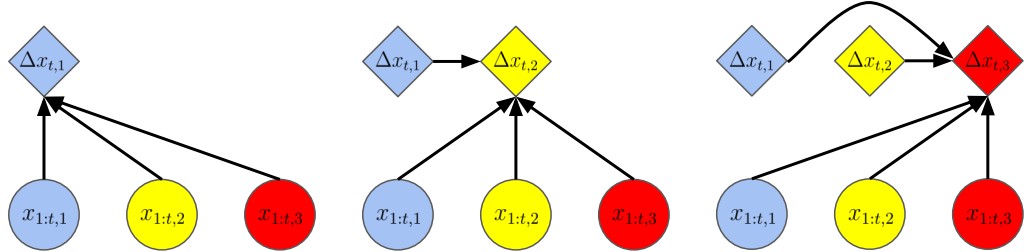

Figure 2: At inference time, a model is not *required* to *simultaneously* generate the trajectories for all of the agents at a specific time step. An alternative strategy is to allow the model to generate the agents' trajectories one at a time, and let the model use the previously generated trajectories to inform the trajectories it generates for the remaining agents.

### 2.2 `baller2vec` is a (conditional) generative model.

`baller2vec` is a recently described *multi-entity* Transformer that can model sequences of *sets* (the underlying data structure for multi-agent spatiotemporal systems), as opposed to sequences of individual inputs (like words in a sentence). When used to model the game of basketball, the input at each time step for `baller2vec` is a set of feature vectors where each feature vector contains information about the identity and location of a player on the court. `baller2vec` maps each input feature vector to an output feature vector, which is then used to "classify" the binned trajectory for that specific player at that specific time step.

Here, we provide a probabilistic interpretation of `baller2vec`, which establishes the theoretical grounds for using the chain rule to generate trajectories agent-wise at each time step in `baller2vec++`. Without loss of generality, we only consider one-dimensional trajectories for a single agent here. To briefly summarize, the outputs of the softmax function over the $n$ binned

trajectories in `baller2vec` can be interpreted as mixture proportions for a mixture of uniform distributions with predetermined bounds that partition the Euclidean trajectory space. Further, because $x_{t+1} = x_t + \Delta x_t$, `baller2vec` is in fact a conditional generative model that assigns a probability to a sequence of trajectories given the initial position of the agent, i.e., $p(\Delta x_1, \Delta x_2, \ldots, \Delta x_T | x_1)$. Using the chain rule, we decompose the joint probability of the trajectories as:

$$p(\Delta x_1, \Delta x_2, \ldots, \Delta x_T) = p(\Delta x_1)p(\Delta x_2|\Delta x_1)\ldots p(\Delta x_T|\Delta x_1, \Delta x_2, \ldots, \Delta x_{T-1})$$

to reflect their temporal structure (we omit the conditional initial position term for brevity). Therefore, new trajectories can be generated from `baller2vec` with the following procedure (see Figure 3):

1. First, sample one of the $n$ different mixture components using the mixture proportions output from the classifier $f$ (i.e., `baller2vec`) conditioned on the agent's current position, i.e., $i \sim \text{Categorical}(\pi_1, \pi_2, \ldots, \pi_n)$ where $[\pi_1, \pi_2, \ldots, \pi_n] = f(x_t)$.

2. Next, sample a trajectory from the uniform distribution associated with the sampled component, i.e., $\Delta x_t \sim \mathcal{U}(a_i, b_i)$.

3. Finally, add the sampled trajectory to the agent's input position to generate the agent's position at the start of the next time step, i.e., $x_{t+1} = x_t + \Delta x_t$.

Let $[\Delta x_{min}, \Delta x_{max})$ be an interval on the real line such that any trajectory $\Delta x < \Delta x_{min}$ or $\Delta x \geq \Delta x_{max}$ has zero density (i.e., such trajectories are humanly impossible). Let $\{[a_i, b_i)\}_{i=1}^n$ be a set of $n$ intervals that partition the interval $[\Delta x_{min}, \Delta x_{max})$ into $n$ bins, i.e., $\cup_{i=1}^n [a_i, b_i) = [\Delta x_{min}, \Delta x_{max})$ and $i \neq j \implies [a_i, b_i) \cap [a_j, b_j) = \emptyset$. Recall that the probability density function (PDF) for a uniform distribution with bounds $-\infty < a < b < \infty$ is:

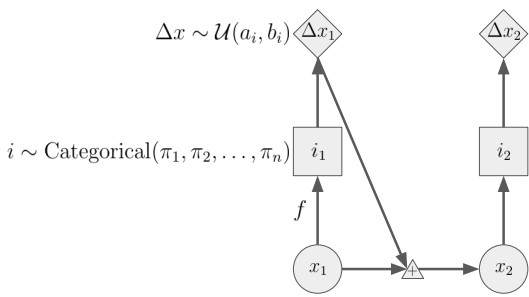

$$p(\Delta x) = \begin{cases} \frac{1}{b-a} & \text{for } \Delta x \in [a, b) \\ 0 & \text{otherwise} \end{cases}$$

Letting $c_i = \frac{1}{b_i - a_i}$, the PDF for a mixture of uniforms with these bounds is thus:

$$p(\Delta x) = \sum_{i=1}^n \pi_i \mathcal{U}(\Delta x; a_i, b_i) = \sum_{i=1}^n \pi_i c_i \quad (1)$$

where $p(\Delta x)$ is the density assigned to $\Delta x$ by the mixture, $\pi_i$ is the mixture proportion for the mixture component indexed by $i$ (i.e., $0 \leq \pi_i \leq 1$ and $\sum \pi_i = 1$), and $\mathcal{U}(\Delta x; a_i, b_i)$ is the density assigned to $\Delta x$ by the uniform distribution with bounds $-\infty < a_i < b_i < \infty$. Because the bounds of the uniform distributions partition $[\Delta x_{min}, \Delta x_{max})$, Equation (1) reduces to:

$$p(\Delta x) = \pi_{i'} c_{i'}$$

where $\Delta x \in [a_{i'}, b_{i'})$ (because the other uniform distributions will assign a density of zero to $\Delta x$). The likelihood for data $D$ (with $|D| = N$) is then:

Figure 3: `baller2vec` can be viewed as a conditional generative model that assigns a probability to a sequence of trajectories given the initial positions of the agents. Here, we show a graphical model depiction of a `baller2vec` model that generates a sequence of one-dimensional trajectories for a single agent. Given the initial position of the agent (the circle containing $x_1$), one of $n$ different uniform distributions (the square containing $i_1$) is sampled using the mixture proportions ($\pi_i$) output by `baller2vec` ($f$). The agent's trajectory (the diamond containing $\Delta x_1$) is then sampled from the selected uniform distribution, which has bounds $-\infty < a_i < b_i < \infty$. At the start of the next time step, the agent's position is $x_2 = x_1 + \Delta x_1$. Maximizing the likelihood of `baller2vec` as a classifier over the binned trajectories is thus equivalent to maximizing its likelihood when assuming the trajectories are generated from a mixture of uniform distributions that partition the Euclidean trajectory space (see Section 2.2 for details).

$$\mathcal{L}(D) = \prod_{j=1}^N p(\Delta x_j) = \prod_{j=1}^N \pi_{j,i'} c_{j,i'}$$

where $\pi_{j,i'}$ is the mixture proportion assigned to the component with $\Delta x_j \in [a_{i'}, b_{i'})$ and $c_{j,i'}$ is the associated density. Taking the negative logarithm of the likelihood gives:

$$-\ln(\mathcal{L}(D)) = -\sum_{j=1}^{N} \ln(\pi_{j,i'}) - \sum_{j=1}^{N} \ln(c_{j,i'}) \tag{2}$$

Because the bounds are fixed, the second summation is a constant, and Equation (2) becomes:

$$-\ln(\mathcal{L}(D)) = -\sum_{j=1}^{N} \ln(\pi_{j,i'}) + C$$

where $C = -\sum_{j=1}^{N} \ln(c_{j,i'})$. Therefore, minimizing the loss of `baller2vec` as a classifier of binned trajectories is equivalent to minimizing the loss of the model when assuming the trajectories are generated from a mixture of uniform distributions as specified in Equation (1).

## 3 Model Architecture

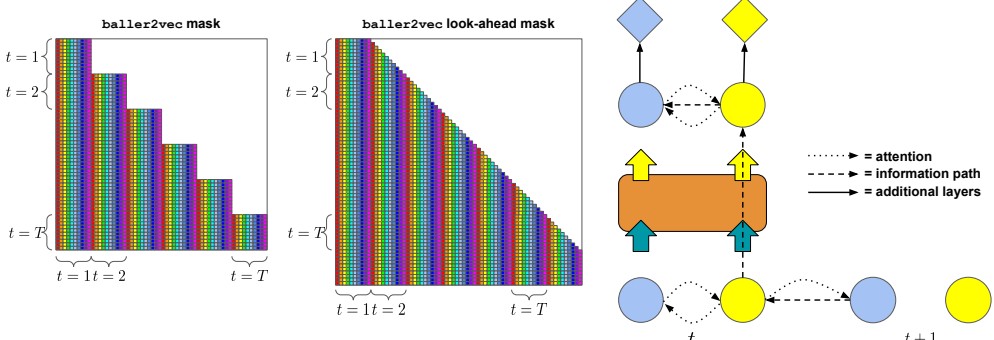

Figure 4: A naive strategy for learning to predict statistically dependent agent trajectories is to adapt the `baller2vec` self-attention mask so that `baller2vec` can "look ahead" at future positions of agents whose trajectories are generated *prior* to the agent being processed in the current time step. However, this look-ahead self-attention mask cannot be used with multi-layer Transformers because doing so necessitates "seeing the future". For example, after the model attends to the blue agent's position at time step $t + 1$ when processing the yellow agent at time step $t$, the yellow agent's resultant feature vector contains information about the blue agent's future position. As a result, when the model attends to the yellow agent while processing the blue agent at the next level, the model is seeing the future.

We motivate our `baller2vec++` architecture by first highlighting an issue that arises in `baller2vec` when trying to model agent trajectories using the chain rule. The `baller2vec` self-attention mask can be adapted so that `baller2vec` "looks ahead" at the future positions of agents whose trajectories are generated prior to the agent being processed in the current time step (Figure 4). However, this look-ahead self-attention mask can only be used with the final layer of the Transformer; otherwise, the model needs to see the future (Figure 4). As a result, `baller2vec` is severely limited in the conditional distribution functions it can learn.

`baller2vec++` (Figure 5) overcomes this limitation by: (1) augmenting the `baller2vec` input with two other sets of feature vectors and (2) using a specially designed self-attention mask. The three sets of feature vectors in `baller2vec++` take the following forms:

1. $z_{t,k} = g_z([e(p_k), x_{t,k}, y_{t,k}, h_{t,k}])$          **(current location information)**

2. $u_{t,k} = g_u([e(p_k), x_{t+1,k}, y_{t+1,k}, h_{t,k}, \Delta x_{t,k}, \Delta y_{t,k}])$     **("look-ahead" information)**

3. $r_k = g_r([e(p_k), x_{1,k}, y_{1,k}, h_{1,k}])$          **(initial location information)**

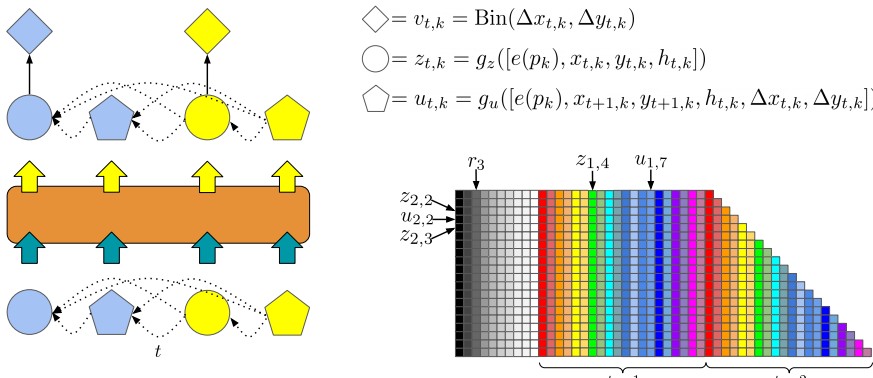

Figure 5: To learn statistically dependent agent trajectories, `baller2vec++` uses a specially designed self-attention mask to simultaneously process three different sets of features vectors in a single Transformer. The three sets of feature vectors consist of location feature vectors like those found in `baller2vec` ($z_{t,k}$), look-ahead trajectory feature vectors ($u_{t,k}$), and starting location feature vectors ($r_k$; not shown). As can be seen in these partial depictions of `baller2vec++` and the `baller2vec++` self-attention mask, this design allows the model to integrate information about *concurrent* agent trajectories through *multiple* Transformer layers without seeing the future.

166 where $g_z$, $g_u$, and $g_r$ are multilayer perceptrons (MLPs), $e$ is an agent embedding layer, and $h_{t,k}$ is a
167 vector of optional contextual features for agent $p_k$ at time step $t$. $z_{t,k}$ is the same location feature
168 vector used in `baller2vec` and contains information about a specific agent's identity and the agent's
169 location at time step $t$. $u_{t,k}$ is a "look-ahead" trajectory feature vector that contains information about
170 a specific agent's identity, the agent's location at the *next time step $t+1$*, and the agent's trajectory
171 at time step $t$, i.e., $(x_{t+1,k} - x_{t,k}, y_{t+1,k} - y_{t,k})$. Lastly, $r_k$ is a starting location feature vector that
172 contains information about a specific agent's identity and the agent's location at time step $t = 1$.
173 The $r_k$ feature vectors are necessary so that `baller2vec++` can "see" the initial locations of *all the*
174 *agents* when processing the agents agent-wise in the first time step.

175 These three sets of feature vectors are combined to form a $(K + 2TK) \times F$ matrix $Z$ such that
176 the first $K$ rows consist of the $K$ $r_k$ feature vectors, and the remaining $2TK$ rows consist of the
177 $TK$ $z_{t,k}$ and $TK$ $u_{t,k}$ feature vectors interleaved with one another, i.e., each $z_{t,k}$ is followed by its
178 corresponding $u_{t,k}$ in the matrix. This matrix is passed into the Transformer along with the specially
179 designed self-attention mask, which encodes the following dependencies (see Figure 5):

1. When processing $r_{k_1}$, `baller2vec++` is exclusively allowed to "look" at each $r_{k_2}$ (i.e.,
   `baller2vec++` cannot look at any location or look-ahead feature vectors when processing
   $r_{k_1}$).

2. When processing $z_{t_2,k_2}$, `baller2vec++` is allowed to "look" at: (i) each $r_{k_1}$, (ii) any $z_{t_1,k_1}$
   where (a) $t_1 < t_2$ **or** (b) $t_1 = t_2$ **and** $k_1 \leq k_2$, and (iii) any $u_{t_1,k_1}$ where (a) $t_1 < t_2$ **or** (b)
   $t_1 = t_2$ **and** $k_1 < k_2$.

3. When processing $u_{t_2,k_2}$, `baller2vec++` is allowed to "look" at: (i) each $r_{k_1}$, (ii) any $z_{t_1,k_1}$
   where (a) $t_1 < t_2$ **or** (b) $t_1 = t_2$ **and** $k_1 \leq k_2$, and (iii) any $u_{t_1,k_1}$ where (a) $t_1 < t_2$ **or** (b)
   $t_1 = t_2$ **and** $k_1 \leq k_2$.

189 Each processed $z_{t,k}$ feature vector is then passed through a linear layer that is followed by a softmax,
190 which gives a probability distribution over the trajectory bins for agent $p_k$ at time step $t$. Similar to
191 `baller2vec`, the loss for each sample is:

$$\mathcal{L} = \sum_{t=1}^{T} \sum_{k=1}^{K} -\ln(f(Z)_{t,2k-1}[v_{t,k}]) \tag{3}$$

192 where $f(Z)_{t,2k-1}[v_{t,k}]$ is the probability assigned to the trajectory bin $v_{t,k}$ (where $v_{t,k} =$
193 $\text{Bin}(\Delta x_{t,k}, \Delta y_{t,k})$ is an integer from one to $n^2$) by $f$, i.e., Equation (3) is the negative log-likelihood
194 (NLL) of the data according to the model.

Because any ordering of a chain rule decomposition of a joint probability produces the same value, e.g.:

$$p(\Delta x_{t,1})p(\Delta x_{t,2}|\Delta x_{t,1})p(\Delta x_{t,3}|\Delta x_{t,1}\Delta x_{t,2}) = p(\Delta x_{t,3})p(\Delta x_{t,2}|\Delta x_{t,3})p(\Delta x_{t,1}|\Delta x_{t,3}\Delta x_{t,2})$$

like [11], we shuffled the order of the agents in each training sequence to encourage the model to learn joint probabilities of the agent trajectories that are approximately permutation invariant with respect to the ordering of the agents.

## 4  Experiments

We tested `baller2vec++` on two different datasets. To highlight the pathological behavior of models that assume agent trajectories are statistically independent at each time step, we trained scaled down versions of `baller2vec++` and `baller2vec` on a toy dataset consisting of simulated trajectories for two perfectly coordinated agents. Additionally, to demonstrate the efficacy of `baller2vec++` in real world settings, we trained `baller2vec++` and `baller2vec` on a dataset of trajectories for professional basketball players.

### 4.1  Toy dataset

Each training sample was initialized with the agents starting at $(-1, 0)$ and $(1, 0)$ on a grid in random order (i.e., the first agent could be placed to either the left or the right of the origin). At each time step, one of nine actions (corresponding to the $3{\times}3$ grid surrounding the agent) was sampled from a uniform distribution, and each of the agents was translated along this trajectory. This process was repeated for 20 time steps (see Figure 6(a) for a sample).

### 4.2  Basketball dataset

We used the same National Basketball Association (NBA) dataset[3] employed by Alcorn and Nguyen [6], whom we paraphrase here:

> The NBA dataset consists of trajectories from 631 games from the 2015-2016 season, which were split into 569/30/32 training/validation/test games, respectively. During training, each sequence was sampled using the following procedure: (1) randomly select a training game, (2) randomly select a starting time from the game, (3) take the following four seconds of data and downsample it to 5 Hz from the original 25 Hz, and then (4) randomly (with a probability of 0.5) rotate the court $180°$. This sampling procedure gave us access to on the order of $\sim$82 million different (albeit overlapping) training sequences. For both the validation and test sets, $\sim$1,000 different, *non-overlapping* sequences were selected for evaluation by dividing each game into $\lceil \frac{1{,}000}{N} \rceil$ non-overlapping chunks (where $N$ is the number of games), and using the starting four seconds from each chunk as the evaluation sequence.

### 4.3  Model

Our `baller2vec++` and `baller2vec` models for the basketball dataset closely followed [6], and so largely resemble the original Transformer architecture [12]. Specifically, the Transformer settings were: $d_{\text{model}} = 512$ (the dimension of the input and output of each Transformer layer), eight attention heads, $d_{\text{ff}} = 2048$ (the dimension of the inner feedforward layers), six layers, no dropout, and no positional encoding. Each MLP (i.e., $g_z$, $g_u$, and $g_r$) had 128, 256, and 512 nodes in its three layers, respectively, and a ReLU nonlinearity following each of the first two layers. The player embeddings [13] had 20 dimensions, and $h_{t,k}$ was a binary variable indicating the side of the frontcourt for player $p_k$ (i.e., the direction of his team's hoop) at time step $t$. Lastly, the 11 ft $\times$ 11 ft 2D Euclidean trajectory space was binned into 121 1 ft $\times$ 1 ft squares.

We used the Adam optimizer [14] with an initial learning rate of $10^{-6}$, $\beta_1 = 0.9$, $\beta_2 = 0.999$, and $\epsilon = 10^{-9}$ to update the model parameters, of which there were $\sim$19 million. The learning rate was

---

[3]`https://github.com/linouk23/NBA-Player-Movements`

reduced to $10^{-7}$ after 20 epochs of the validation loss not improving. Models were implemented in PyTorch and trained on a single NVIDIA GTX 1080 Ti GPU for ~650 epochs (seven days) where each epoch consisted of 20,000 training samples, and the validation set was used for early stopping.

For the toy dataset, we used scaled down versions of the basketball models with $d_{\text{model}} = 128$, four attention heads, $d_{\text{ff}} = 512$, and two layers in the Transformer. Additionally, each MLP had two layers with 64 and 128 nodes, respectively. The models were trained for 50 epochs of 500 samples per epoch (~10.5 minutes) using a single learning rate of $10^{-5}$.

## 5 Results

### 5.1 `baller2vec++` can effectively model coordinated agents in both simulated and real settings

For the toy dataset, the training loss for `baller2vec` converged to $\sim 2.2 \approx -\ln(\frac{1}{9})$, i.e., the model was simply independently guessing the trajectories for both agents at every time step. In contrast, the training loss for `baller2vec++` converged to $\sim 1.1 \approx -\ln(\frac{1}{9}) \div 2$, which is the expected loss for a model that perfectly learns the deterministic relationship between the agents' trajectories (because the prediction for the second agent will always contribute $-\ln(1.0) = 0$ to the loss).

When generating trajectories with `baller2vec`, the agents are completely uncoordinated, with each agent following an independent random walk around the grid (Figure 6(b)). In contrast, trajectories generated by `baller2vec++` display the same coordinated agent behavior as the training data (Figure 6(c)).

For the basketball dataset, `baller2vec++` achieved an average NLL of 0.472 on the test set, 8.9% better than the average NLL for

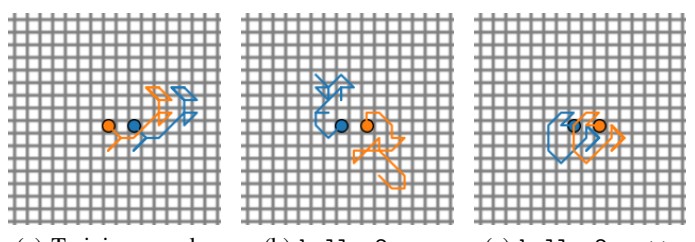

(a) Training sample.  (b) `baller2vec`.  (c) `baller2vec++`.

Figure 6: When trained on a dataset of perfectly coordinated agent trajectories (a), the trajectories generated by `baller2vec` are completely *uncoordinated* (b) while the trajectories generated by `baller2vec++` are perfectly coordinated (c). Animated versions can be found in the code repository.

`baller2vec` (0.518) (see Figure S1 for trajectories generated by `baller2vec++` and `baller2vec`). As was observed in [6], the trajectory bin distributions for `baller2vec` become much more certain after observing a portion of the sequence (Figure 7), which suggests `baller2vec` may be inferring some of the shared hidden variables (e.g., plays) influencing the players. If that hypothesis was true, the performance gap between `baller2vec++` and `baller2vec` should be largest at the beginning of the sequence (before any shared hidden variables can be inferred by `baller2vec`). Indeed, the average NLL for `baller2vec++` in the *first time step* of each test set sequence (1.567) is 16.1% better than the average NLL for `baller2vec` (1.869), while the average NLL for `baller2vec++` in the *last time step* of each test set sequence (0.420) is only 9.7% better than the average NLL for `baller2vec` (0.465) (see Figure 7).

### 5.2 `baller2vec++` makes better predictions when conditioned on concurrent trajectory information from other agents

Implicit in much of our discussion has been the intuition that providing a model with additional (relevant) information will improve its performance. To empirically test this conjecture, we compared the performance of `baller2vec++` when predicting the trajectory of a specific basketball player placed in the *first position* of the player order (i.e., when $k = 1$) vs. predicting the trajectory for that *same player* placed in the *last position* (i.e., when $k = 10$). Specifically, for each player in each test sequence, we calculated the NLL of the player's trajectory in the first time step[4] with the player in the

---

[4]Because, as previously discussed, the benefits of `baller2vec++` were most pronounced in the first time step.

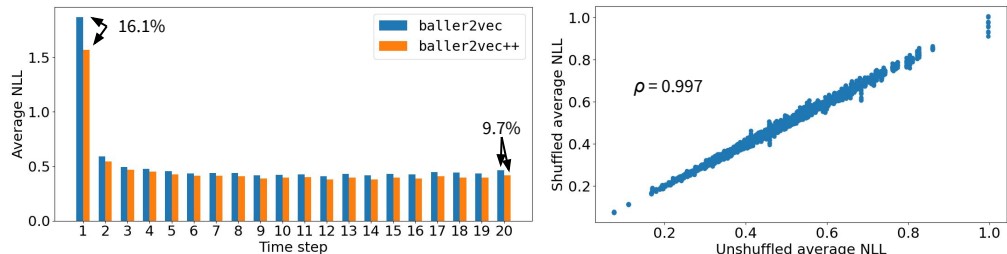

Figure 7: **Left**: when modeling the trajectories of professional basketball players, the performance gap between `baller2vec++` and `baller2vec` is largest at the beginning of the sequence, before shared unobserved variables can be inferred by `baller2vec`. Each bar indicates a model's average NLL over the entire test set for that particular time step. For full sequences, `baller2vec++` outperforms `baller2vec` by 8.9%. **Right**: the joint probability assigned to a sequence by `baller2vec++` is approximately permutation invariant with respect to the order of the agents. For each point, its $x$ value indicates `baller2vec++`'s average NLL for a test set sequence using the *original* order of the agents in the sequence, while its $y$ value indicates `baller2vec++`'s average NLL *for the same sequence* with the order of the agents *shuffled*. The shuffled average NLLs are highly correlated with their corresponding unshuffled average NLLs.

*first position* of the player order. Next, we moved the player to the *last position* of the player order, and then randomly shuffled the remaining nine players 10 times, calculating the NLL for the player in the last position each time. Finally, we calculated the average percent change in the last position NLLs relative to their corresponding first position NLLs. On average, moving a player from the first to the last position improved the NLL for the player's trajectory by 14.6%.

## 5.3 The joint probability assigned to a sequence by `baller2vec++` is approximately permutation invariant with respect to the order of the agents

To determine whether or not `baller2vec++` respects the fact that any ordering of a chain rule decomposition of a joint probability produces the same value, we measured how much the average NLL for each test sequence in the basketball dataset varied when the order of the agents changed. Specifically, for each test set sequence, we shuffled the order of the agents 10 times. Then, for each permuted sequence, we calculated the percent error[5] in the average NLL relative to the original, *unshuffled* sequence. Across all test sequences, the average percent error was only $\pm 1.5\%$. Further, as can be seen in Figure 7, the shuffled average NLLs are highly correlated with their corresponding unshuffled average NLLs (Pearson correlation coefficient = 0.997), i.e., the joint probability assigned to a sequence by `baller2vec++` is approximately permutation invariant with respect to the order of the agents.

## 6 Conclusion and Future Work

In this paper, we have shown how the commonly used independence assumption of many multi-agent spatiotemporal models can severely limit their ability to learn to emulate coordinated agents. By relaxing this independence assumption in `baller2vec`, `baller2vec++` was able to more accurately model the trajectories of professional basketball players. Models for other multi-agent spatiotemporal environments, such as pedestrian traffic (see [15] for a survey) and vehicle traffic (e.g., [16–19]), may also benefit from the look-ahead approach used by `baller2vec++`. Additionally, the interleaved input design of `baller2vec++` could be useful when modeling other systems involving many entities interacting through time, such as social media platforms (e.g., [20, 21]). However, confronting the quadratic complexity of the Transformer attention mechanism as the number of entities grows large in these datasets is an open problem, but recent work in sparse Transformers (e.g., [22–27]) shows encouraging progress.

---

[5]See: `https://en.wikipedia.org/wiki/Approximation_error#Formal_Definition`.

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
