# OpenReview forum: "baller2vec++: A Look-Ahead Multi-Entity Transformer For Modeling Coordinated Agents"
_NeurIPS.cc/2021/Conference — NeurIPS 2021 Submitted_

### Official Review · Reviewer_yWcc · 2021-07-12

**Rating:** 5
**Confidence:** 3

**Summary:**

The authors proposed a multi-entity Transformer that can effectively model coordinated agents called baller2vec++. The proposed method applies a specially designed self-attention mask to a mixture of location and “look-ahead” trajectory sequences to learn the distributions of statistically dependent agent trajectories. Unlike the conventional baller2vec, the proposed method can learn to emulate the behavior of coordinated agents in a simulated toy dataset. Moreover, the proposed method modeled the trajectories of professional basketball players more accurately than the conventional baller2vec.

The strength of this paper seems to be 1) the experimental results outperforming the conventional baller2vec as described above, 2) relaxing the independence assumption of agents in baller2vec and adding “look ahead” information to the transformer input, and 3) the joint probability assigned to a sequence was approximately permutation invariant with respect to the order of the agents by the post-hoc analysis.


**Ethics Review Area:**

["I don’t know"]

**Limitations And Societal Impact:**

Limitations were addressed. Societal impact was not addressed but the reason was described.

**Main Review:**

However, there are two major weaknesses of this paper: 1) comparison only with the conventional baller2vec. Since trajectory prediction is a common task in machine learning, a comparison with other approaches (e.g., RNN-based) is required. 2) methodologically minor change (as described above 2 of the strength) and small theoretical contribution. Although relaxing the independence assumption of agents is an interesting idea, the modification of the previous model seems to be incremental in total. The theoretical ground for the permutation invariant will increase the value of this paper, in addition to the post-hoc analysis. For these reasons, I consider this paper does not meet the criteria for acceptance.

Minor comments:

33 i.e., the macro-intent for one agent does not depend on the macro-intents of the other agents. Macro-intents in [Zhan et al. 2019] may be shared among agents. The trajectory prediction among agents was not shared as the authors mentioned.

Figure 4 is confusing between baller2vec and baller2vec++. Did baller2vec use  “look ahead” information?

250 for the toy dataset, mainly training loss was evaluated. Please specify the reason (i.e., why not test loss) and whether the loss is training/test loss in the following sentences. For example, “254 (because the prediction for the second agent will always contribute − ln(1.0) = 0 to the loss).” is unknown whether training/test loss.

In Figure 6, was it better to predict different directions in the left figure (proposed method) from the ground truth? Moreover, I did not understand why the prediction loss for the second agent would always contribute − ln(1.0) = 0 in this case.


**Time Spent Reviewing:**

5 hours

---

> ### Author Response · Authors · 2021-08-06
> **Regarding Our Evaluation Experiments and Contributions**
>
> Thank you for your review and for highlighting the strengths of our manuscript! We will address your other comments below.
>
> **comparison only with the conventional baller2vec. Since trajectory prediction is a common task in machine learning, a comparison with other approaches (e.g., RNN-based) is required**
>
> We will add a sentence noting that `baller2vec` outperformed both a graph recurrent neural network and a single agent Transformer by wide margins (10.5% and 18.0%, respectively) on the same NBA dataset in Alcorn and Nguyen (2021), which is why we focused on comparing `baller2vec++` to `baller2vec` in our manuscript.
>
> **methodologically minor change (as described above 2 of the strength) and small theoretical contribution. Although relaxing the independence assumption of agents is an interesting idea, the modification of the previous model seems to be incremental in total.**
>
> We do not believe our extension to `baller2vec` (i.e., integrating parallel sequences of future positions) was a priori obvious. Additionally, an important contribution of our work is drawing attention to the commonly used agent-independence assumption of multi-agent spatiotemporal models and the issues that can arise from that assumption. As far as we are aware, we are the first to highlight the pathological behavior that can result from models using the agent-independence assumption.
>
> **Macro-intents in [Zhan et al. 2019] may be shared among agents. The trajectory prediction among agents was not shared as the authors mentioned.**
>
> Can you clarify what you mean here? The full architecture of Zhan et al. (2019) consists of two recurrent neural networks: (1) a trajectory prediction network and (2) a macro-intents prediction network (as found in the authors’ implementation [here](https://github.com/ezhan94/multiagent-programmatic-supervision/blob/a1d9152d4c8a287474953cba093c28fef2a05979/models/macro_vrnn.py#L101), which we linked to in a footnote). The *shared* macro-intent for the agents at each time step in the trajectory prediction network is the concatenation of the *predicted* individual macro-intents, which are conditionally statistically independent as revealed through their implementation.
>
> **Figure 4 is confusing between baller2vec and baller2vec++. Did baller2vec use “look ahead” information?**
>
> Figure 4 is highlighting the “seeing the future” issue that arises in `baller2vec` when using a naive look-ahead self-attention mask to attempt to model statistically dependent agent trajectories. `baller2vec` does not use look-ahead information. The `baller2vec++` architecture depicted in Figure 5 includes pentagons that represent the parallel “look-ahead” sequences, which, together with the different self-attention mask design, distinguishes it from `baller2vec`.
>
> **250 for the toy dataset, mainly training loss was evaluated. Please specify the reason (i.e., why not test loss) and whether the loss is training/test loss in the following sentences. For example, “254 (because the prediction for the second agent will always contribute − ln(1.0) = 0 to the loss).” is unknown whether training/test loss.**
>
> There is no distinction between the training loss and the test loss here because the dataset was randomly generated on the fly. That is, the final training loss of our model is effectively its test loss.
>
> **In Figure 6, was it better to predict different directions in the left figure (proposed method) from the ground truth?**
>
> Figure 6 is meant to highlight how `baller2vec++` (right) can learn to model the coordinated behavior of the agents in the toy dataset while `baller2vec` (middle) cannot. We intentionally labeled the left figure “Training sample” as opposed to “Ground truth” to convey that the models were not supposed to be *predicting* the agent trajectories depicted in the left figure. By design, our dataset is fundamentally unpredictable (because the agents move in a random direction at each time step), but it is possible to model the coordinated behavior of the agents using a model like `baller2vec++` that learns the statistical dependencies of the agent trajectories.
>
> **Moreover, I did not understand why the prediction loss for the second agent would always contribute − ln(1.0) = 0 in this case.**
>
> Because the two agents are perfectly coordinated, knowing the trajectory of one agent at a specific time step removes any uncertainty about the trajectory for the other agent at that time step. For example, if the first agent moves to the left at a specific time step, then the probability that the second agent will also move to the left at that time step is 1.0. Therefore, a model that perfectly learned to emulate this system should assign a probability to the situation where both agents move to the left of $p(\text{left}, \text{left}) = p(\text{left}) p(\text{left}|\text{left}) = \frac{1}{9} 1.0 = \frac{1}{9}$.

---

### Official Review · Reviewer_VCp8 · 2021-07-15

**Rating:** 3
**Confidence:** 2

**Summary:**

Summary:
The paper proposed baller2vec++, utilizes transformer architecture for modeling multi-agent trajectories with dependencies. Applying a specially designed self-attention mask, baller2vec++ outperforms baller2vec on professional basketball dataset.

**Limitations And Societal Impact:**

Yes

**Main Review:**

Strength:
1. The paper is clearly written.

Weakness:
1. The paper seems missing a related work section.

2. The experiments is not convincing. Despite a toy dataset, only a Basketball dataset experiment is provided.

3. The only baseline is baller2vec in the experiments.

4. It resembles baller2vec and the novelty seems to be questionable.

**Time Spent Reviewing:**

1.5

---

> ### Author Response · Authors · 2021-08-06
> **Regarding Our Evaluation Experiments**
>
> Thank you for your review and for highlighting the clarity of our manuscript. We will address your other comments below.
>
> **The paper seems missing a related work section.**
>
> We combined the introduction and related work into a single “Introduction and Related Work” section because it seemed to work best for the flow of the paper.
>
> **The experiments is not convincing. Despite a toy dataset, only a Basketball dataset experiment is provided.**
>
> Because the behavior of basketball players is quite complex and coordinated, we believe it is an excellent dataset for demonstrating the utility of our architecture. Unfortunately, the commonly used pedestrian datasets (i.e., ETH, HOTEL, UNIV, ZARA1, and ZARA2) are *extremely* small, which makes them insufficient for evaluating an architecture that is known to thrive on large datasets (researchers designing Transformers for language modeling are not expected to evaluate their architectures on datasets only containing thousands of sentences). Specifically, whereas our NBA dataset has ~82 million different time steps in the training set, the total number of time steps for all of the previously mentioned pedestrian datasets combined is only 6,982. Likewise, the [H3D traffic dataset](https://usa.honda-ri.com/H3D) (Patil et al., 2020) [1] only contains 27,721 time steps.
>
> We also note that Social-BiGAT (Kosaraju et al., 2019), a NeurIPS 2019 paper [2], was exclusively evaluated on the previously mentioned pedestrian datasets, and Zhan et al., (2019), an ICLR 2019 paper [3], exclusively evaluated their model on an NBA dataset, so there is precedent in the community to evaluate multi-agent architectures in a single domain. However, we look forward to the opportunity to evaluate `baller2vec++` on other large, multi-agent datasets when they become available.
>
> **The only baseline is baller2vec in the experiments.**
>
> We will add a sentence noting that `baller2vec` outperformed both a graph recurrent neural network and a single agent Transformer by wide margins (10.5% and 18.0%, respectively) on the same NBA dataset in Alcorn and Nguyen (2021), which is why we focused on comparing `baller2vec++` to `baller2vec` in our manuscript.
>
> **It resembles baller2vec and the novelty seems to be questionable.**
>
> We do not believe our extension to `baller2vec` (i.e., integrating parallel sequences of future positions) was a priori obvious. Additionally, an important contribution of our work is drawing attention to the commonly used agent-independence assumption of multi-agent spatiotemporal models and the issues that can arise from that assumption. As far as we are aware, we are the first to highlight the pathological behavior that can result from models using the agent-independence assumption.
>
> **References**
>
> [1] Abhishek Patil, Srikanth Malla, Haiming Gang, and Yi-Ting Chen. The h3d dataset for full-surround 3d multi-object detection and tracking in crowded urban scenes. In International Conference on Robotics and Automation, 2019.
>
> [2] Vineet Kosaraju, Amir Sadeghian, Roberto Martín-Martín, Ian Reid, Hamid Rezatofighi, and Silvio Savarese. Social-bigat: Multimodal trajectory forecasting using bicycle-gan and graph attention networks. In Advances in Neural Information Processing Systems, 2019.
>
> [3] Eric Zhan, Stephan Zheng, Yisong Yue, Long Sha, and Patrick Lucey. Generating multi-agent trajectories using programmatic weak supervision. In International Conference on Learning Representations, 2019.

---

### Official Review · Reviewer_K2zL · 2021-07-16

**Rating:** 5
**Confidence:** 4

**Summary:**

This work presents an extension to the existing baller2vec transformer architecture. In particular, a new attention mask is used internally that is lower-triangular (offset by one block diagonal) rather than block-lower-triangular. This enables baller2vec++ to attend over entities at both previous and one-step future timesteps (in particular by attending over one-step predictions). The method was demonstrated in two experimental scenarios, an illustrative toy example and a dataset of NBA basketball trajectories.

**Limitations And Societal Impact:**

The authors have not adequately addressed the limitations and potential negative societal impact of their work, these sections are not present in the paper.

**Main Review:**

Strengths:
- Compared to baller2vec, this work presents a novel extension to the self-attention mask which factors in one-step predictions and allows for statistical dependency between agents (which is not seen in prior works). Further, while permutation-variance might be an issue, the authors show in Section 5.3 that such a concern does not pan out empirically, which is reassuring.

Weaknesses:
- There are a significant amount of missing baselines in the experimental work. It is understood that this work is an extension of another, however this paper should be completely standalone, even if that requires bringing forward experiments and comparisons from the predecessor work.
    - The same criticism applies to the related work, which is missing many references and relies on the reader having an intimate knowledge of the predecessor paper.

- The experimental suite is limited to a simple illustrative scenario and data from NBA basketball games. However, in order to make claims such as the Section 5.1 heading, one would need to evaluate baller2vec++ on more datasets covering a wider range of scenarios (NBA basketball play is arguably at one end of the human motion spectrum). This is especially important to verify if the novel self-attention mask is necessary for modeling every day behavior such as human driving or walking within crowds, or if it is only empirically useful in a highly-coordinated and dynamic sport such as basketball.

**Time Spent Reviewing:**

2

---

> ### Author Response · Authors · 2021-08-06
> **Regarding Prior Work and Evaluation Experiments**
>
> Thank you for your review and for highlighting the strengths of our manuscript! We will address your other comments below.
>
> **There are a significant amount of missing baselines in the experimental work. It is understood that this work is an extension of another, however this paper should be completely standalone, even if that requires bringing forward experiments and comparisons from the predecessor work.**
>
> We will add a sentence noting that `baller2vec` outperformed both a graph recurrent neural network and a single agent Transformer by wide margins (10.5% and 18.0%, respectively) on the same NBA dataset in Alcorn and Nguyen (2021), which is why we focused on comparing `baller2vec++` to `baller2vec` in our manuscript.
>
> **The same criticism applies to the related work, which is missing many references and relies on the reader having an intimate knowledge of the predecessor paper.**
>
> Can you please provide specific examples of missing references? We tried to include the multi-agent modeling papers that were most relevant to our work (i.e., specifically addressing coordinated behavior) or would serve as good starting points in a literature survey. The multi-agent modeling field is very large, so, unfortunately, some relevant work will always be omitted given the space constraints.
>
> **The experimental suite is limited to a simple illustrative scenario and data from NBA basketball games. However, in order to make claims such as the Section 5.1 heading, one would need to evaluate baller2vec++ on more datasets covering a wider range of scenarios (NBA basketball play is arguably at one end of the human motion spectrum). This is especially important to verify if the novel self-attention mask is necessary for modeling every day behavior such as human driving or walking within crowds, or if it is only empirically useful in a highly-coordinated and dynamic sport such as basketball.**
>
> Unfortunately, the commonly used pedestrian datasets (i.e., ETH, HOTEL, UNIV, ZARA1, and ZARA2) are *extremely* small, which makes them insufficient for evaluating an architecture that is known to thrive on large datasets (researchers designing Transformers for language modeling are not expected to evaluate their architectures on datasets only containing thousands of sentences). Specifically, whereas our NBA dataset has ~82 million different time steps in the training set, the total number of time steps for all of the previously mentioned pedestrian datasets combined is only 6,982. Likewise, the [H3D traffic dataset](https://usa.honda-ri.com/H3D) (Patil et al., 2020 [1]) only contains 27,721 time steps.
>
> We also note that Social-BiGAT (Kosaraju et al., 2019), a NeurIPS 2019 paper [2], was exclusively evaluated on the previously mentioned pedestrian datasets, and Zhan et al. (2019), an ICLR 2019 paper [3], exclusively evaluated their model on an NBA dataset, so there is precedent in the community to evaluate multi-agent architectures in a single domain. If the primary issue is that our language is too strong, we are happy to adjust our wording. Otherwise, we look forward to the opportunity to evaluate `baller2vec` on other large, multi-agent datasets when they become available.
>
> **References**
>
> [1] Abhishek Patil, Srikanth Malla, Haiming Gang, and Yi-Ting Chen. The h3d dataset for full-surround 3d multi-object detection and tracking in crowded urban scenes. In International Conference on Robotics and Automation, 2019.
>
> [2] Vineet Kosaraju, Amir Sadeghian, Roberto Martín-Martín, Ian Reid, Hamid Rezatofighi, and Silvio Savarese. Social-bigat: Multimodal trajectory forecasting using bicycle-gan and graph attention networks. In Advances in Neural Information Processing Systems, 2019.
>
> [3] Eric Zhan, Stephan Zheng, Yisong Yue, Long Sha, and Patrick Lucey. Generating multi-agent trajectories using programmatic weak supervision. In International Conference on Learning Representations, 2019.

---

> > ### Comment · Reviewer_K2zL · 2021-08-31
> > **Response to Authors**
> >
> > Thank you for your response!
> >
> > A (fantastic) reference for prior works on multi-agent modeling is: "Human Motion Trajectory Prediction: A Survey" by Rudenko et al. There are 100+ references within that review paper alone, not to mention the related work sections of other recent works such as "AgentFormer" in ICCV 2021.
> >
> > The authors correctly point out that commonly-used pedestrian datasets are very small, which is especially true for ETH/UCY (which are well-known in the field). However, it's a bit strange to see only H3D referenced in the author's response rather than the significantly more popular nuScenes, Argoverse, or Lyft Level 5 datasets. Lyft Level 5 in particular has 40+ million time steps!
> >
> > Since this work lacks evaluation on public datasets and other reviewers have identified the same issue, I will maintain my score.

---

> > > ### Author Response · Authors · 2021-09-09
> > > **Related Work/Datasets in baller2vec++ vs. the AgentFormer**
> > >
> > > **However, it's a bit strange to see only H3D referenced in the author's response rather than the significantly more popular nuScenes, Argoverse, or Lyft Level 5 datasets. Lyft Level 5 in particular has 40+ million time steps!**
> > >
> > > Thank you for pointing us to those datasets. We were admittedly not familiar with popular traffic datasets because we do not do research on traffic/autonomous driving problems.
> > >
> > > **A (fantastic) reference for prior works on multi-agent modeling is: "Human Motion Trajectory Prediction: A Survey" by Rudenko et al.**
> > >
> > > We agree and we did in fact cite Rundeko et al. (2020).
> > >
> > > **not to mention the related work sections of other recent works such as "AgentFormer" in ICCV 2021.**
> > >
> > > The AgentFormer (Yuan et al., 2021) is an interesting comparison case to `baller2vec++` for several reasons.
> > >
> > > First, we feel the Related Work section of Yuan et al. (2021) further highlights the challenges of choosing which prior research to include in the extremely limited amount of space that is available for conference papers. Specifically, we cite:
> > >
> > > * Chandra et al. (2019)
> > > * Chang et a al. (2019)
> > > * Deo and Trivedi (2018)
> > > * Felsen et al. (2018)
> > > * Yeh et al. (2019)
> > > * Zhan et al. (2019)
> > > * Zheng et al. (2016)
> > >
> > > which were all omitted from Yuan et al. (2021). Further, Yuan et al. (2021) (which was accepted to ICCV after `baller2vec++` was submitted to NeurIPS) does not cite `baller2vec`, which was submitted to arXiv nearly two months prior to the AgentFormer (2021-02-05 vs. 2021-03-25) and is highly relevant.
> > >
> > > Second, other than the small pedestrian datasets, the only dataset used in Yuan et al. (2021) was nuScenes, which only has 1,000 scenes * 20 seconds per scene * 2 Hz = 40,000 time steps of data.
> > >
> > > Third, the AgentFormer [had an error in its implementation that led to a data leak during evaluation](https://github.com/Khrylx/AgentFormer/issues/5), which resulted in [the AgentFormer's performance being overstated](https://github.com/Khrylx/AgentFormer/issues/5#issuecomment-915180574). The error was only able to be identified because the authors thankfully released their code. An important contribution of our work to the community is that the code for our entire pipeline, from data generation to model training, is public, which distinguishes it from many other papers that use sports datasets like Yeh et al. (2019) and Felsen et al. (2018). We feel the lack of standardized benchmarks for sports datasets, combined with the general lack of open source code, significantly complicates comparing different models and has hindered the pace of research in this domain.
> > >
> > > Lastly, the AgentFormer also makes the conditional independence assumption that was the focus of our manuscript, which we feel further emphasizes the importance of documenting the implications of this assumption for the community.

---

### Official Review · Reviewer_GRxC · 2021-08-06

**Rating:** 3
**Confidence:** 5

**Summary:**

The authors construct a framework based on multi-entity Transformers to handle the group dynamics (i.e. dependence between agents at each time step.) They implement a “look ahead” strategy to improve performance and coordination of agents.  This approach produces results much better than baller2vec, the model's predecessor.  The results are demonstrated on a basketball dataset and a toy set with perfect coordination.  While the performance on these datasets is good, there are a number of concerns regarding the paper including an absence of comparison to other methods, dependence on a fixed/"binnable" output space, only approximate permutation invariance, and ability to handle variable or a high number of entities.  As a result, the impact of this approach appears very limited.


**Limitations And Societal Impact:**

The authors state their work does not introduce any social challenges, which seems reasonable.

The authors speak to some limitations, but this has not been adequately addressed nor explored via experiments.  See comments above around the discrete output space, and number/noise/variability of the number of agents.

**Main Review:**

The approach in the present work produces improved results over a past model (baller2vec) on a basketball dataset.  However, the significance of the work is severely limited by the lack of comparison to other methods and demonstrated performance on other datasets especially where there is less coordination and the number of entities may be large, variable, or noisy.  As currently formulated, the paper is more of an application and minimal extension of the Transformer framework to a very domain-specific problem which comes with strong assumptions (fixed entities, small discretizable spatial extent) that enables the approach to work.  Therefore the contribution to the broader domain of multi-agent trajectory forecasting appears severely limited.

The overall text is clear enough to read.  However, I would suggest to the authors to spend more of the body of the paper focused on conducting experiments and comparisons to other approaches and ablation studies (knocking out players, adding poor data) as opposed to derivations of dependent probabilities pertaining to a past method.

No physical measure (i.e. in terms of meters) is provided.  Given that this is about predicting agent's motion, this is a critical oversite.   While the model is probabilistic in nature, a physical measure of performance still needs to be provided in addition to NLL.  If you're trying to model motion to handle whether two pedestrians (or a pedestrian and a car) are going to collide, physical distance matters in addition to NLL of a given path.  A top-k measure in RMSE is needed.

The model is only tested on a single “real world” dataset plus a toy dataset with perfect coordination, both with a fixed number of entities who are present in every frame.  While this is true in a sports setting (or at least basketball- not true in football where a player may be sent off), this is not true in the general multi-agent setting.  For the work to be more impactful, its performance on other common multi-agent datasets should be demonstrated.  Note, the issue is not that a basketball/sport dataset was used or the primary focus- sport is a tremendous domain for studying multi-agent motion- it is simply that additional analysis in other domains was not performed.  If the author's believe the only real application space for this method is within sport where these assumptions hold, but that this approach is so much better than all other methods for this particular domain, then they should reformulate the paper to express this (both in terms of text and in comparison to other methods).

The fact that the model is only approximately permutation invariant is highly problematic.  Many approaches out there (specifically those based on GNNs) are explicitly permutation equivariant.  While being approximately invariant may be acceptable in a sports setting, it could lead to large errors in more “natural” settings (like pedestrian detection) where movements are less coordinated and the number of agents is large and variable.

The work is largely based on the baller2vec framework, which is a preprint.  The authors spend significant time discussing the “probabilistic interpretation” of this model which is fairly obvious given that it is largely an implementation of the transformer architecture.

The “binned trajectory” approach that is used here dramatically reduces the applicability and usefulness of the approach in any setting outside of the sports domain where the spatial domain is fixed.

The authors discuss the issue with the quadratic dependence on the number of entities in a Transformer framework, but they should also speak to variable number of entities.

How does the model handle bad/missing data?  If there is a gap (or error) in a track, can the model still perform?

There are no comparisons to other SOTA approaches (other than baller2vec).  Comparison to these other methods is critical, especially provided that baller2vec is only a preprint.  These other methods offer advantages in terms of handling variable/missing players, speed, personalization, and other features common in multi-agent task.  Social Bi-GAT is mentioned in the intro, but not compared to.

The statement in 22-24 that these models treat the trajectories as independent is untrue.  Something like Social-LSTM naturally contains information about past states.  Felsen 2018 explicitly conditioned on past trajectories to predict future motion.  The statement that the macro-intents in Zhan et al. are independent is not quite true either.  Although the dependency is not explicitly modeled, because their ordering is based on the group-structure evolution, there is an implicit dependence of the past motions of different agents impacting the other agents (i.e. the role assigned to a given agent, which impacts the prediction of the macro-intent, is very strongly influenced by the past motions of other remaining agents).

In the opening the authors discuss the coordination of agents being driven by a shared unobserved variable, like a play in basketball.  If this is the case, it should be possible to extract and perform some kind of analysis on this meta-state.  If the unobserved variable is changed, but the incoming trajectories are largely fixed (which is common at the start of many different plays), how will the subsequent trajectories change?  Probing this through ablation studies seems very important given the authors' claims.  On the flipside, what happens in less coordinated settings (like pedestrian tracking) where there is no explicit coordination?

**Time Spent Reviewing:**

3.5

---

> ### Author Response · Authors · 2021-08-07
> **Clarifying Our Contributions**
>
> **However, the significance of the work is severely limited by the lack of comparison to other methods and demonstrated performance on other datasets especially where there is less coordination and the number of entities may be large, variable, or noisy.**
>
> We will add a sentence noting that `baller2vec` outperformed both a graph recurrent neural network and a single agent Transformer by wide margins (10.5% and 18.0%, respectively) on the same NBA dataset in Alcorn and Nguyen (2021), which is why we focused on comparing `baller2vec++` to `baller2vec` in our manuscript.
>
> Given that our model is theoretically capable of learning *any* joint distribution of trajectories, we would not expect a dataset featuring *less* coordinated agents to pose a challenge for our architecture.
>
> Additionally, unfortunately, the commonly used pedestrian datasets (i.e., ETH, HOTEL, UNIV, ZARA1, and ZARA2) are *extremely* small, which makes them insufficient for evaluating an architecture that is known to thrive on large datasets (researchers designing Transformers for language modeling are not expected to evaluate their architectures on datasets only containing thousands of sentences). Specifically, whereas our NBA dataset has ~82 million different time steps in the training set, the total number of time steps for all of the previously mentioned pedestrian datasets combined is only 6,982. Likewise, the [H3D traffic dataset](https://usa.honda-ri.com/H3D) (Patil et al., 2020 [1]) only contains 27,721 time steps.
>
> **As currently formulated, the paper is more of an application and minimal extension of the Transformer framework to a very domain-specific problem which comes with strong assumptions (fixed entities, small discretizable spatial extent) that enables the approach to work. Therefore the contribution to the broader domain of multi-agent trajectory forecasting appears severely limited.**
>
> We do not believe our extension to `baller2vec` (i.e., integrating parallel sequences of future positions) was a priori obvious. Additionally, an important contribution of our work is drawing attention to the commonly used agent-independence assumption of multi-agent spatiotemporal models and the issues that can arise from that assumption. As far as we are aware, we are the first to highlight the pathological behavior that can result from models using the agent-independence assumption. Lastly, we do not believe our binning approach is in any way a limitation. It is simply a modeling choice.
>
> **No physical measure (i.e. in terms of meters) is provided. Given that this is about predicting agent's motion, this is a critical oversite. While the model is probabilistic in nature, a physical measure of performance still needs to be provided in addition to NLL. If you're trying to model motion to handle whether two pedestrians (or a pedestrian and a car) are going to collide, physical distance matters in addition to NLL of a given path. A top-k measure in RMSE is needed.**
>
> RMSE is not a more valid performance metric than the NLL metric we reported in the manuscript. We note that the other NeurIPS reviewers did not express similar concerns about our chosen performance metric. If a user wished to estimate the probability of a collision using our model (or any multimodal model for that matter, e.g., a mixture of Gaussians), the way they would do that is by running many simulations and calculating the proportion of simulations that resulted in a collision.
>
> **The model is only tested on a single “real world” dataset plus a toy dataset with perfect coordination, both with a fixed number of entities who are present in every frame. While this is true in a sports setting (or at least basketball- not true in football where a player may be sent off), this is not true in the general multi-agent setting. For the work to be more impactful, its performance on other common multi-agent datasets should be demonstrated. Note, the issue is not that a basketball/sport dataset was used or the primary focus- sport is a tremendous domain for studying multi-agent motion- it is simply that additional analysis in other domains was not performed. If the author's believe the only real application space for this method is within sport where these assumptions hold, but that this approach is so much better than all other methods for this particular domain, then they should reformulate the paper to express this (both in terms of text and in comparison to other methods).**
>
> We note that Social-BiGAT (Kosaraju et al., 2019), a NeurIPS 2019 paper [2], was exclusively evaluated on the previously mentioned pedestrian datasets, and Zhan et al., (2019), an ICLR 2019 paper [3], exclusively evaluated their model on an NBA dataset, so there is precedent in the community to evaluate multi-agent architectures in a single domain. If the primary issue is that our language is too strong, we are happy to adjust our wording. Otherwise, we look forward to the opportunity to evaluate `baller2vec++` on other large, multi-agent datasets when they become available.
>
> **The fact that the model is only approximately permutation invariant is highly problematic. Many approaches out there (specifically those based on GNNs) are explicitly permutation equivariant. While being approximately invariant may be acceptable in a sports setting, it could lead to large errors in more “natural” settings (like pedestrian detection) where movements are less coordinated and the number of agents is large and variable.**
>
> Prior GRNN work like Yeh et al. (2019) is only permutation equivariant because they use the overly strong agent independence assumption that is the focus of our manuscript. As we stated above, we would not expect a dataset featuring *less* coordinated agents to pose a challenge for our architecture.
>
> **The authors spend significant time discussing the “probabilistic interpretation” of this model which is fairly obvious given that it is largely an implementation of the transformer architecture.**
>
> The purpose of our probabilistic derivation was to show that a binned trajectory output is equivalent to modeling a continuous trajectory output under certain assumptions.
>
> **The “binned trajectory” approach that is used here dramatically reduces the applicability and usefulness of the approach in any setting outside of the sports domain where the spatial domain is fixed.**
>
> Can you clarify what you see is the relationship between our output representation, which is modeling trajectory *deltas*, and the nature of the “spatial domain”? As we stated above, we do not believe our binning approach is in any way a limitation. It is simply a modeling choice. If required by the application, our proposed architecture can model the trajectory deltas using a mixture of Gaussians, but this modeling choice poses its own challenges, such as potentially allowing a player to move much farther than physically possible in a single time step.
>
> **The authors discuss the issue with the quadratic dependence on the number of entities in a Transformer framework, but they should also speak to variable number of entities.**
>
> There are a number of potential solutions for handling a variable number of entities depending on the exact nature of the dataset; for example, the maximum number of entities ever seen in a single time step could always be used in the input, and sequences containing fewer entities could be filled in with “dummy” feature vectors.
>
> **How does the model handle bad/missing data? If there is a gap (or error) in a track, can the model still perform?**
>
> We did not test the missing data case, but there are a number of potential solutions depending on the exact nature of the dataset; for example, randomly replacing an agent’s identity embedding with a special “missing” embedding during training.
>
> (continued below)

---

> > ### Author Response · Authors · 2021-08-07
> > **Clarifying Our Contributions (continued)**
> >
> > **There are no comparisons to other SOTA approaches (other than baller2vec). Comparison to these other methods is critical, especially provided that baller2vec is only a preprint. These other methods offer advantages in terms of handling variable/missing players, speed, personalization, and other features common in multi-agent task. Social Bi-GAT is mentioned in the intro, but not compared to.**
> >
> > Because our dataset is ~800x larger than other NBA datasets, e.g., Felsen et al. (2018) [1] and Zhan et al. (2019) [2], and even larger than pedestrian datasets, scaling up these architectures to ensure a fair comparison would require a considerable hyperparameter search. Further, Yeh et al. (2019) did not include the hyperparameters of their architecture in their paper, did not publicly release their code, and did not reply to an email request for code/model hyperparameters, which made direct comparison to their approach impossible. While Felsen et al. (2018) did include their neural network’s hyperparameters in their paper, code for their “role-alignment” procedure is not available. Unlike Yeh et al. (2019), Felsen et al. (2018), and many other papers that use sports datasets, the code for our entire pipeline, from data generation to model training, is public, which is an important contribution of our work to the community. We feel the lack of standardized benchmarks for sports datasets, combined with the general lack of open source code, significantly complicates comparing different models and has hindered the pace of research in this domain.
> >
> > **The statement in 22-24 that these models treat the trajectories as independent is untrue. Something like Social-LSTM naturally contains information about past states. Felsen 2018 explicitly conditioned on past trajectories to predict future motion. The statement that the macro-intents in Zhan et al. are independent is not quite true either.**
> >
> > Lines 22-24 read:
> >
> > >However, nearly all multi-agent spatiotemporal models (e.g., [1-6]) implicitly (through their loss functions) assume the trajectories of the agents at each time step are statistically *independent* given the agents' previous locations (Figure 1).
> >
> > We believe we were quite precise here, and we included Figure 1 to make our motivation even more explicit. Additionally, *all* of the predicted trajectory outputs in Felsen et al. (2018) are conditionally statistically independent of one another because they are only derived from the input trajectories. Lastly, the full architecture of Zhan et al. (2019) consists of two recurrent neural networks: (1) a trajectory prediction network and (2) a macro-intents prediction network (as found in the authors’ implementation [here](https://github.com/ezhan94/multiagent-programmatic-supervision/blob/a1d9152d4c8a287474953cba093c28fef2a05979/models/macro_vrnn.py#L101), which we linked to in a footnote). The *shared* macro-intent for the agents at each time step in the trajectory prediction network is the concatenation of the *predicted* individual macro-intents, which are conditionally statistically independent as revealed through their implementation.
> >
> > **If the unobserved variable is changed, but the incoming trajectories are largely fixed (which is common at the start of many different plays), how will the subsequent trajectories change? Probing this through ablation studies seems very important given the authors' claims.**
> >
> > Can you provide a concrete way to “change” an unobserved variable in the way you are suggesting here? The play, along with any other unobserved variables, are modeled internally by the Transformer by applying the attention mechanism across players and across time to our parallel location and look-ahead sequences. The Transformer learns to model these unobserved variables by being optimized to model the (assumption-free) joint distribution of the trajectories through the chain rule of probability. If the initial trajectories of different sequences are truly the same/”similar”, then we would expect them to have similar distributions of future trajectories.
> >
> > **References**
> >
> > [1] Abhishek Patil, Srikanth Malla, Haiming Gang, and Yi-Ting Chen. The h3d dataset for full-surround 3d multi-object detection and tracking in crowded urban scenes. In International Conference on Robotics and Automation, 2019.
> >
> > [2] Vineet Kosaraju, Amir Sadeghian, Roberto Martín-Martín, Ian Reid, Hamid Rezatofighi, and Silvio Savarese. Social-bigat: Multimodal trajectory forecasting using bicycle-gan and graph attention networks. In Advances in Neural Information Processing Systems, 2019.
> >
> > [3] Eric Zhan, Stephan Zheng, Yisong Yue, Long Sha, and Patrick Lucey. Generating multi-agent trajectories using programmatic weak supervision. In International Conference on Learning Representations, 2019.

---

### Public Comment · ~Michael_A._Alcorn1 · 2021-11-20
**Datasets in Other Multi-Agent Trajectory Papers Accepted to NeurIPS 2021**

For posterity, I want to note that at least two other multi-agent trajectory modeling papers that were accepted to NeurIPS this year only evaluated their methods on a single, large real-world dataset. "[GRIN: Generative Relation and Intention Network for Multi-agent Trajectory Prediction](https://openreview.net/forum?id=ephWA7KaWmD)" evaluated their method on a small simulated dataset (50K training sequences) and a preprocessed NBA dataset (100K training sequences). "[Collaborative Uncertainty in Multi-Agent Trajectory Forecasting](https://openreview.net/forum?id=sO4tOk2lg9I)" evaluated their method on the nuScenes dataset (1,000 scenes as mentioned below) and Argoverse (206K training sequences). Additionally, the 2020 NeurIPS paper "[Learning Agent Representations for Ice Hockey](https://proceedings.neurips.cc/paper/2020/hash/d90e5b6628b4291225cba0bdc643c295-Abstract.html)" was only evaluated on a hockey dataset.

---

### Decision · Program_Chairs · 2021-09-27

**Decision:**

Reject

**Comment:**

All ratings were "reject". The key problems the reviewers found with this paper is that it appears to be very close to baller2vec, the experimental comparisons to other models are not broad enough, there are insufficient ablation experiments, and generally there's a sense that the contribution is not substantial enough to warrant publication.

The authors were, however, very thorough and attentive in their rebuttal and discussion, addressing various reviewers' concerns as best as they could. While I felt this weighed in the authors' favor, it wasn't enough to sway the reviewers or me, and I feel rejection is still the correct decision.

I encourage the authors to follow up on some of the more involved suggestions the reviewers made, test on a wider range of datasets to assess how general the method is, try to distinguish this work a little more from baller2vec, etc. If the results improve, I believe the paper will be more viable in the future.